# Factors affecting the maximum outcome payments of social impact bonds

Huan Wang [1,2], Naipeng Chao[1], Jiaxi Chen[2]*, Mengqi Chen[2], Tingting Fu[2]

**1** School of Media and Communication, Shenzhen University, Shenzhen, China, **2** Research Center for Political Party Development, Shenzhen Reform and Opening up Executive Leadership Academy, Shenzhen, China

* chenjiaxi@szroela.org.cn

## Abstract

As governments globally seek market-based solutions to address complex social challenges, Social Impact Bonds (SIBs) have gained prominence in public policy reforms for their potential to align fiscal accountability with social innovation. SIBs are innovative financial tools designed to fund social projects through outcome-based payments, lower government financial pressure, and increase the efficiency with which social problems are solved. In a SIB, the maximum outcome payment refers to the highest amount the outcome payer is willing to provide, and is one of the key factors attracting investor participation. It is typically calculated as the sum of service costs and fiscal savings. However, the current approach often prioritizes reducing government expenditure, without sufficiently accounting for the risks and interests of investors. This improper formulation of maximum outcome payments may cause financing bottleneck and hinder the sustainable development of SIBs. To achieve a balance between government cost control and investor returns, this paper systematically examines the key factors influencing maximum outcome payments through multiple regression analysis, with a focus on macroeconomic factors and bond characteristics. The results indicate that maximum outcome payments are correlated with inflation, capital raised, and the size of the target population for the SIB. Therefore, when determining maximum outcome payments, issuers should consider not only fiscal savings but also bond characteristics and macroeconomic factors.

## Introduction

Whether environmental pollution, health, poverty, employment, or something else, numerous countries are being faced with increasingly complex social problems that governments, investors, and social organizations need to collaborate together to solve. Social impact bonds (SIBs) are a multi-sectoral social financing mechanism dedicated to helping the government solve social problems, putting social benefits first when measuring the impact of project completion, and taking into account

**Data availability statement:** All relevant data are within the paper and its Supporting information files.

**Funding:** This work was supported by China Postdoctoral Science Foundation (Certificate Number: 2024M752096).

**Competing interests:** The authors have declared that no competing interests exist.

economic benefits [1]. Between the issuance of the first SIB in the UK, in 2010, and June 2024, 295 SIBs had been issued with a total size of about 764m USD according to the GO Lab Impact Bond Dataset [2]. This rapid adoption has sparked intense academic and policy debates, crystallizing around three dominant paradigms: reformist perspectives emphasizing public sector innovation [3,4], financial models exploring sustainable investment frameworks [5–7], and critical examinations of marketization effects [8–10]. These competing narratives reflect both the transformative potential of SIBs and the implementation challenges they face as they transition from experimental pilots to mainstream policy instruments.

SIBs are an outcomes-based financing method, which transfers financial risks related to the pursuit of public goals to private investors [11]. The return on investment depends on the degree to which the target outcome is achieved. If the objectives are not achieved, investors receive neither a return nor repayment of principal [8]. Consequently, SIBs have inherent problems of uncertain yield and high risk. According to the GO Lab Impact Bond Dataset, most SIBs publish maximum outcome payments at the time of issuance. The maximum outcome payment refers to the highest possible amount that the outcome payer commits to pay to investors if all agreed-upon outcomes are fully achieved [5]. This cap serves both as a risk containment measure for the payer and an outcome-based incentive for investors. However, the definition and application of maximum outcome payment may vary across projects and jurisdictions. For example, the maximum outcome payment in the UK's HMP Peterborough SIB was explicitly tied to reductions in recidivism among short-sentenced prisoners, with predefined thresholds and payment rates [12]. Conversely, in the Utah High Quality Preschool Program SIB in the United States, the payment was based on the number of children avoiding special education placement, with a fixed per-child payment that accrued up to a maximum cap [13].

Existing studies on maximum outcome payments of SIBs have mainly focused on qualitative research, but there are few relevant quantitative studies. Gustafsson-Wright et al. elaborated the methods of setting the maximum outcome payment [5]. They point out that the government, as the outcome payer, publishes rate cards that define the maximum payment for each outcome type. Hulse et al. summarized average maximum payments for 12 SIB projects across seven countries, but their sample was small [14]. Olson et al. found differences in SIB programs between the UK and US for their target population, capital raised, and maximum outcome payment, but their scope was limited [15]. La Torre et al. identified a close relationship between the expected public savings and the maximum outcome payment [16]. According to the GO Lab Impact Bond Dataset and OECD Working Papers, the maximum outcome payment is calculated as the sum of service costs and fiscal savings, determined through validated administrative data and control trials. However, the existing method often focuses more on the goal of reducing government expenditure, while insufficiently considering the interests of investors and the risks they bear [16,17]. Improved methods are needed to balance government cost and investor returns.

The Arbitrage Pricing Theory (APT) proposed by Ross pointed out that the returns of financial assets can be determined driven by macroeconomic factors and specific factors relating to those assets [18,19]. APT proposes a multifactor pricing model that has been used in many studies examining the determinants of bond yields [20]. Notably, both internal and external factors are attached to bonds, which can affect bond returns [21]. Such research provides empirical evidence on the internal and macroeconomic variables that determine bond returns.

In terms of macroeconomic variables, the interest rate with universal reference value in the financial market is the benchmark interest rate [22], which other financial products can refer to when determining their own interest rates [23]. Scholars and practitioners have regarded the risk-free treasury rate as the benchmark interest rate [24]. Accordingly, the expected return of a bond can be estimated from the risk-free treasury rate. In addition, inflation rate can be regarded as a proxy for the quality of economic management [25]. Kurniasih and Restika estimated the impact of interest rates, exchange rates, inflation, and foreign ownership on bond yields [26]. Their empirical results showed that inflation has a positive impact on bond yields. In addition, Tekula & Andersen proved that inflation has a positive effect on government bond yields [27].

In terms of bond characteristics, scholars have mainly studied influencing factors such as bond period and capital raised. An evaluation of the term structure of interest rates revealed the relationship between bond price and maturity, namely that the longer the term, the higher the interest rate [28,29]. Che-Yahya et al. conducted a study on 61 companies that issued bonds in the Malaysian bond market in 2012 and concluded that bonds with longer maturities would have higher price volatility and investors would be compensated with higher yields [30]. Thus, bond maturity has a positive impact on returns.

The maximum outcome payment in SIBs is a crucial factor for attracting investors, calculated as the sum of service costs and fiscal savings. Currently, this payment primarily focuses on reducing government expenditure without sufficiently accounting for the risks and interests of investors, which may cause financing bottlenecks and hinder the sustainable development of SIBs. This study analyzes the factors influencing maximum outcome payments from macro perspectives and bond characteristics to ensure a balance between government costs and investor returns. Unlike previous qualitative studies, this work uses an empirical approach with a global sample from GO Lab Impact Bond Dataset, making results more representative. Key findings reveal that inflation, capital raised, and target population size significantly influence maximum outcome payments. This research helps issuers set reasonable payments, attracting more investors and promoting SIB development, while ensuring investors bear appropriate risks and receive fair returns.

## Research design

### Theoretical framework

This paper integrates APT, Theory of Change, and public management theories to systematically explain SIB pricing mechanisms. The model establishes macroeconomic variables and bond characteristics from APT as the foundational pricing layer, transforms social value through Theory of Change's "input-output-outcome" chain, and ultimately balances stakeholder interests via public management theory. These three tiers form a progressive relationship of financial pricing→social adjustment→policy regulation, simultaneously explaining empirical findings while informing policy design, and reveal SIBs' unique logic as a market-society-government tripartite coupling mechanism.

### Arbitrage pricing theory

APT provides the foundational logic for SIB pricing by identifying macroeconomic variables (e.g., inflation, treasury rates) and bond-specific characteristics (e.g., capital raised, maturity) as determinants of financial returns [19,21]. Unlike traditional bonds, SIBs incorporate APT's multifactor framework to quantify risks tied to macroeconomic volatility (e.g., currency fluctuations) and project-specific parameters (e.g., target population size). This approach ensures that payments

reflect both systemic risks (e.g., inflation-indexed adjustments) and idiosyncratic factors (e.g., capital commitment tiers), aligning investor compensation with measurable financial and social uncertainties. By adapting APT's risk-return calculus to outcome-based contracts, SIBs bridge market-driven pricing with social impact objectives, offering a structured method to balance fiscal constraints and investor incentives.

### Theory of change

Theory of Change bridges financial pricing with social impact by mapping the causal pathway from inputs (e.g., investor capital) to outcomes (e.g., reduced recidivism) [31]. It postulates a cause-and-effect or input-outcome sequence to guide action, and specifies relevant indicators and formative and summative performance measures [9]. This layer explains why traditional financial variables may lack significance in SIBs: Returns are tied to verified social outcomes rather than market liquidity or credit risk. For instance, a SIB targeting youth employment may tie payments to verified job placements, with pricing adjusted for program scalability. By embedding the Theory of Change model into contract design, SIBs transform social value creation into quantifiable financial metrics, ensuring the payments reflect achieved impact rather than speculative market conditions.

### Outcomes-driven public management

SIBs are outcomes-based contractual mechanisms for financing government programs. Outcomes-driven public management describes an orientation towards "results-focused" strategies that aim to improve the performance and effectiveness of government [32]. Distinctive features of outcomes-based public management systems include performance measures that focus on project outcomes rather than inputs, outputs, or processes [33]. These systems feature the linking of outcome goals across levels of government and service delivery partners and the design of new mechanisms for focusing management attention on the specified outcomes. In so doing, they balance stakeholder interests through dynamic payment structures (e.g., tiered payments linked to capital commitments) and institutional safeguards (e.g., disclosure requirements and tax incentives).

### Research hypothesis

Based on the above analysis of existing research, we formulated and tested the following null hypotheses, illustrated in S1 Fig.

Hypothesis 1 (H1). There is a positive correlation between treasury rate and maximum outcome payment.

Treasury bonds can be regarded as risk-free bonds because there is no difference in expected returns [34]. Scholars and practitioners have regarded the treasury rate as the benchmark interest rate [23,24], which constitutes a universal reference value in the financial market. Other financial products can determine their own interest rates by referring to this benchmark rate [23,35]. Therefore, the expected return of a bond can be estimated from the risk-free treasury bond rate. As benchmark interest rates of the SIBs investigated here, this paper selects treasury rates having identical issue date and period as the corresponding SIBs. It is assumed that the maximum outcome payments of SIBs are positively correlated with treasury rates.

Hypothesis 2 (H2). There is a positive correlation between inflation rate and maximum outcome payment.

The inflation rate can be regarded as a proxy for the quality of economic management. The higher the inflation rate, the lower the government's credit rating. A large number of empirical studies have proven that bonds with lower credit ratings have higher yields [36], which finding is also supported by Nanayakkara & Colombage, who argue that high inflation requires higher return on earnings [37]. Research by Campbell showed that bond returns are related to the expected inflation rate over the life of the bond [38]. Campbell and Ammer likewise suggested that changes in inflation rates may alter bond returns [38]. Kurniasih and Restika estimated the impact of interest rates, exchange rates, inflation, and foreign

ownership on bonds between 2010 and 2013, and found that inflation has a positive impact on bond yields [26]. Simoski further highlighted that government bond yields are mainly determined by inflation [39]. For present purposes, the consumer price index (CPI) is selected as the proxy variable for inflation rate. A positive correlation is assumed between CPI and the maximum outcome payments of SIBs.

Hypothesis 3 (H3). There is a positive correlation between bond period and maximum outcome payment.

The term structure of interest rates defines a relation between the yield of a bond and its maturity [40]. The yield to maturity of medium-term bonds is higher than that of short-term bonds [41], and when the maturity of the issue is longer, the liquidity will be weakened. Issuers must attract investors by raising the yield, and bond investors will demand higher risk compensation for the bonds with longer maturities. Che-Yahya et al. conducted a study on 61 companies that issued bonds in the Malaysian bond market in 2012 and concluded that bonds with longer maturities would have higher price volatility and investors would be compensated with higher yields [30]. Accordingly, bond maturity has a positive impact on returns. In this paper, it is assumed that bond period is positively correlated with the maximum outcome payment of a SIB.

Hypothesis 4 (H4). There is a positive correlation between capital raised and maximum outcome payment.

Existing studies have highlighted that when issuance scale is large, the corresponding debt repayment risk will increase, which will eventually lead to an increase in the bond interest rate [42]. SIBs are an outcome-based payment financing mechanism without margin [43], and investors are at risk of loss of principal [44]. Accordingly, the investors with higher investments should be given correspondingly higher payments. In this paper, it is assumed that the maximum outcome payment of a SIB is positively correlated with the capital raised.

Hypothesis 5 (H5). There is a negative correlation between size of the target population and maximum outcome payment.

Unlike bonds that pay principal and interest at maturity, SIBs are a financing method that pays according to results, and there is no guarantee that investors will eventually get the principal [45,46]. The returns to investors instead depend on how much of the target population that achieves the targeted outcome [9]. As such, when analyzing factors that influence the maximum outcome payments of SIBs, we must consider the target population as one such factor. SIBs are not paid based on input or output, but rather on outcome, and the size of the target population can affect the quality of service and the outcome. Therefore, this paper assumes that the size of the target population is negatively correlated with the maximum outcome payment of a SIB.

**Variables and data**

**Variables.** The maximum outcome payment of a SIB is the explained variable. Considering the literature and the actual situation concerning SIBs, this paper included two levels of explanatory variables in the empirical model: macro factors and bond factors. Macro influence factors include treasury rate and inflation rate, while bond characteristics include bond period, capital raised, and size of target population. Variable descriptions are detailed in Table 1.

(1) Benchmark interest rates. This paper selects treasury rates having the same issue date and period as corresponding SIBs to serve as the benchmark interest rates. Treasury rate is denoted as TR.

(2) Inflation rate. The inflation rate can be regarded as a proxy for the quality of economic management. CPI is selected as the proxy variable for inflation in this paper.

(3) Bond period. The period spanning from the start date of service provision to the end of a SIB project, denoted as BP.

(4) Capital raised. The total capital raised by a SIB, which represents its financing scale. SIBs are published in 39 different countries, but the capital raised is uniformly given in millions of dollars. The capital raised is denoted as CR.

**Table 1. Variables and descriptions.**

| Variable Category | Name | Description | Unit |
|---|---|---|---|
| Explained variable | MP | Maximum outcome payment | million dollars |
| Macro influence factors | TR | Treasury rate with the same issue date and bond period as the corresponding SIB | % |
| | CPI | Inflation as measured by the consumer price index | % |
| Bond characteristics | BP | Bond period, from the start date of service provision to the end of the project | years |
| | CR | Capital raised by a SIB | million dollars |
| | TP | Size of the target population | thousand persons |

(5) Size of target population. SIB projects differ considerably in the size of their target populations, which is given in units of thousands of persons. Target population size is denoted as TP.

**Data sources.** This paper performed an empirical analysis on 295 SIBs issued from 2010 to 2024. Data on the maximum outcome payment, bond period, capital raised and size of target population were obtained from the Government Outcomes Lab created as a partnership between the University of Oxford and the UK Government. Missing data were supplemented from the gray literature, which includes the website of each SIB and its report. To ensure data quality and comparability, all gray literature sources were subject to a triangulation process involving cross-verification with multiple independent sources, prioritization of government or evaluator-backed documents, and consistency checks. In addition, treasury rates in each country having the same maturity as the selected SIBs were sourced from the Investing database, and CPI data from the Express Wealth Management database. We screened each bond issued and excluded the bonds that lack maximum outcome payments, maturity, capital raised, a treasury rate with the same maturity, and had abnormal data. After screening and processing, 118 SIBs were included in the quantitative analysis of this paper.

Table 2 summarizes the investigated SIBs by issuing country. It can be concluded that among all the countries issuing SIBs, the United States, India, the United Kingdom, and Australia have relatively high information transparency. Other countries issued SIBs that were all excluded as samples. These countries need to increase the transparency of information disclosure concerning their SIBs. Moreover, the included SIBs account for only 40% of all SIBs issued, indicating that the information transparency of SIBs issued globally needs to be improved in general.

## Descriptive statistical analysis

The descriptive statistics of included SIBs are shown in Table 3. From these statistics, the following conclusions can be drawn.

The descriptive statistics of the 118 SIB samples revealed significant variations across variables. The mean, standard deviation, maximum, and minimum of maximum outcome payments were respectively 7.42m USD, 0.76m USD, 40.15m USD, and 0.15m USD. The maximum outcome payments of SIBs vary greatly. When the outcome payer estimates the maximum outcome payment, it takes into account influence on the target group, the expenditure of the outcome payer, and the broad interests of society. This also indicates that the social impacts of the various SIB projects are quite different.

For macroeconomic factors, the mean value, standard deviation, maximum value, and minimum value of treasury rates were respectively 1.18%, 0.14%, 7.82%, and −0.63%. Treasury rates in this study had issue dates and periods that matched the corresponding SIBs. While the 118 SIBs were issued between 2010 and 2024 and came from 16 countries, the overall treasury rate of the samples did not fluctuate much. The mean value, standard deviation, maximum value, and minimum value of CPI were respectively 1.65%, 0.09%, 4.91%, and −0.63%. This indicates that the CPI of the 16 countries did not fluctuate much during the past 10 years.

**Table 2. Number of issued and included SIBs.**

| Country | Issued SIBs | Included SIBs | Included bond proportion in issued SIBs |
|---|---|---|---|
| United States | 28 | 22 | 79% |
| India | 4 | 3 | 75% |
| United Kingdom | 99 | 60 | 61% |
| Australia | 15 | 8 | 53% |
| Republic of Korea | 2 | 1 | 50% |
| Israel | 2 | 1 | 50% |
| Canada | 9 | 4 | 44% |
| France | 10 | 4 | 40% |
| Germany | 3 | 1 | 33% |
| Colombia | 3 | 1 | 33% |
| New Zealand | 3 | 1 | 33% |
| Portugal | 23 | 6 | 26% |
| South Africa | 4 | 1 | 25% |
| Finland | 4 | 1 | 25% |
| Belgium | 4 | 1 | 25% |
| Netherlands | 18 | 4 | 22% |
| Other countries | 23 | 0 | 0% |
| Total | 295 | 118 | 40% |

**Table 3. Descriptive statistical analysis results.**

| Variable | Mean | Std. Deviation | Maximum | Minimum |
|---|---|---|---|---|
| MP (million dollars) | 7.42 | 0.76 | 40.15 | 0.15 |
| TR (%) | 1.18 | 0.14 | 7.82 | −0.63 |
| CPI (%) | 1.65 | 0.09 | 4.91 | −0.63 |
| BP (year) | 4.64 | 0.27 | 30.00 | 1.00 |
| CR (million dollars) | 3.16 | 0.40 | 25.00 | 0.07 |
| TP (thousand persons) | 15.30 | 7.68 | 633 | 0.01 |

Regarding bond characteristics, the average value, standard deviation, maximum value, and minimum value of the bond period were respectively 4.64 years, 0.27 years, 30 years, and 1 year. Thus, the maturities of SIBs vary widely. It may be that bond maturity is affected by the amount of financing and the ease with which a target social problem can be solved. The mean value, standard deviation, maximum value, and minimum value of bond financing were respectively 3.16m USD, 0.40m USD, 25m USD, and 0.07m USD. The capital raised with SIBs thus varies widely. The mean value, standard deviation, maximum value, and minimum value of the targeted population were respectively 15.30 thousand persons, 7.68 thousand persons, 633 thousand persons, and 10 persons. This shows that the sizes of the target groups receiving assistance through SIB projects varies greatly.

## Multiple regression analysis

**Collinearity analysis.** In a multiple regression model, independent variables that are strongly and linearly related may result in inaccurate estimation results. Therefore, the multicollinearity of independent variables should be tested when multiple regression models are conducted. The variance inflation factor (VIF) can be utilized to detect multicollinearity. Generally, a VIF larger than 10 indicates that severe multicollinearity may significantly influence the regression estimation

results [47]. As shown in Table 4, the VIF values of all variables in this multiple linear regression model are less than 10, so there is no multicollinearity problem.

**Heteroscedasticity test.** The problem of inaccurate results due to heteroscedasticity is easy to exist in regression analysis. Therefore, in this analysis, heteroscedasticity is first tested. The results of heteroscedasticity test are shown in Table 5. The significance of White test is less than 0.05, indicating that heteroscedasticity exists in regression, so heteroscedasticity processing is required.

## Regression results

The multiple regression model used here is as follows:

$$MP_{i,t} = \beta_0 + \beta_1 TR_{i,t} + \beta_2 CPI_{i,t} + \beta_3 BP_{i,t} + \beta_4 CR_{i,t} + \beta_5 TP_{i,t} + \varepsilon_{i,t} \tag{1}$$

In this model, $\beta_0$ is a constant term, $\beta_1$, $\beta_2$,…, $\beta_5$ are the regression coefficients, $\varepsilon_{it}$ is the random error term, $i$ is country and $t$ is time. The error term includes the social impact converted into money, as well as other factors not taken into account.

In order to eliminate the error caused by country and time, individual effects and time effects were controlled for country and time in this analysis. In addition, the above analysis proved the existence of heteroscedasticity, so heteroscedasticity robust regression was adopted. It can be seen from Table 6 that the adjusted R2 of the multiple linear regression equation is 0.68, indicating that the explanation level of each variable in the multiple linear regression equation to the maximum outcome payment is 0.68, which is a high degree of explanation. The above variables were used to conduct regression analysis on the maximum outcome payment, and the multiple regression results were shown in the following table.

As can be seen from Table 6, TR and BP has no significant effect on MP. At the significant level of 1%, CPI has a significant positive effect on MP, and the regression coefficient is 2.62889; at the significant level of 1%, CR has a significant positive effect on MP, and the regression coefficient is 1.07588, while TP has a significant negative effect on MP at the significance level of 5%, and the regression coefficient is −0.00161. The regression coefficients plot for determinants of maximum outcome payments is shown in S2 Fig. The results are consistent with the regression findings, indicating that CPI, CR, and TP have a significant effect on MP.

Therefore, after removing the insignificant variable TR and BP, the regression analysis was conducted again, the results of which are presented in Table 7.

After removing the insignificant variable TR and BP, all other variables are still significant. The final model is as follows:

**Table 4. Variance Inflation Factor (VIF) test results.**

| Variable | VIF |
|---|---|
| TR | 1.34 |
| CPI | 1.36 |
| BP | 2.1 |
| CR | 1.68 |
| TP | 1.68 |
| Mean VIF | 1.63 |

**Table 5. Results of heteroscedasticity test.**

| Metric | chi2(20) | Prob > chi2 |
|---|---|---|
| Value | 88.81 | 0.000 |

Table 6. Multiple regression results.

| | (1) MP | (2) MP | (3) MP | (4) MP | (5) MP | (6) MP |
|---|---|---|---|---|---|---|
| TR | 1.45155*** (3.05) | | | | | 0.40945 (1.26) |
| CPI | | 4.71027*** (8.62) | | | | 2.62889*** (5.70) |
| BP | | | 1.33767*** (5.72) | | | 0.32802 (1.51) |
| CR | | | | 1.42104*** (12.18) | | 1.07588*** (8.32) |
| TP | | | | | 0.00165* (1.83) | −0.00161** (−2.39) |
| _cons | 5.67517*** (2.02) | −0.02979 (5.66) | 1.15551 (3.79) | 2.97918*** (−0.18) | 7.12566*** (7.77) | −1.84706* (2.72) |
| N | 118 | 118 | 118 | 118 | 118 | 118 |
| r2 | 0.074 | 0.391 | 0.220 | 0.561 | 0.028 | 0.690 |
| r2_a | 0.07 | 0.39 | 0.21 | 0.56 | 0.02 | 0.68 |

*$p<0.1$, ** $p<0.05$, *** $p<0.01$.

Table 7. Multiple regression results after excluding the explanatory variables.

| | (1) model 1 |
|---|---|
| CPI | 2.84801*** (6.45) |
| CR | 1.16899*** (10.17) |
| TP | −0.00093* (−1.69) |
| _cons | −0.57759 (−0.75) |
| N | 118 |
| r2 | 0.682 |
| r2_a | 0.67 |

*$p<0.1$, ** $p<0.05$, *** $p<0.01$.

$$MP_{i,t} = -0.57759 + 2.84801CPI_{i,t} + 1.16899CR_{i,t} - 0.00093TP_{i,t} \quad (2)$$

## Endogeneity test

Considering the possibility of endogeneity, this paper chooses Heckman two-stage method to solve this problem. In the first stage model, dummy variables are constructed from the median of MP, which is 1 above the median and 0 otherwise. The Probit model was used to regression the dummy variables, and Inverse Mill's Ratio (IMR) was calculated according to the regression results. In the second stage model, IMR was brought into the main effect model as a control variable for re-regression, and the regression results were shown in Table 8. After IMR was added, the significance of each variable

**Table 8. Endogeneity test regression results.**

|  | (1)<br>model 1 | (2)<br>model 2 |
|---|---|---|
| TR | 0.40945<br>(1.26) | 0.46265<br>(1.45) |
| CPI | 2.62889***<br>(5.70) | 1.86833***<br>(3.29) |
| BP | 0.32802<br>(1.51) | 0.22038<br>(1.01) |
| CR | 1.07588***<br>(8.32) | 0.98251***<br>(7.34) |
| TP | −0.00161**<br>(−2.39) | −0.00144**<br>(−2.16) |
| Imr |  | −0.72811**<br>(−2.22) |
| _cons | −1.84706*<br>(−1.70) | 1.81652<br>(0.93) |
| N | 118 | 118 |
| r2 | 0.690 | 0.704 |
| r2_a | 0.68 | 0.69 |

* $p < 0.1$, ** $p < 0.05$, *** $p < 0.01$.

was consistent with that before IMR was added, and IMR was significant, indicating that the results of this paper were still valid after controlling possible endogeneity problems.

## Robustness test

In order to ensure the reliability of the research conclusion, this paper adopts the method of adding independent variables to conduct robustness test. SIBs are mainly issued by governments [43]. As a test of the robustness of the model, macro factors are added as independent variables, namely government revenue. Government revenue is expressed as the ratio of fiscal revenue (FR) to GDP. According to the regression results shown in Table 9, CPI, CR and TP are still significant, indicating that the model has good robustness.

## Results analysis

### Analysis of macro factors

The multiple regression revealed that the treasury rate, as the benchmark interest rate, is not a statistically significant factor. This result refutes Hypothesis 1 (H1) and shows that there is no significant correlation between the treasury rate and maximum outcome payments. The fundamental distinction between SIBs and traditional bonds lies in their risk pricing logic: Traditional bond returns depend on issuer creditworthiness and market interest rate fluctuations, whereas SIBs tie returns entirely to third-party-verified achievement of social outcomes (e.g., crime rate reduction or employment rate improvement) through "outcome-based payment" contracts, forming a "results-driven" pricing model [48]. The core reason for the insignificance of treasury rates in this framework is that SIBs reconstruct the value assessment system through risk transfer. Their payment models are grounded in the causal effects of social interventions (such as net impacts validated by randomized controlled trials) rather than financial market interest rate parameters [49]. Concurrently, domination of the investor structure by philanthropic capital and the social reputation premium compensation mechanism [50] further diminish the benchmark role of risk-free rates in pricing.

**Table 9. Robustness test.**

| | (1) model 1 | (2) model 2 | (3) model 3 | (4) model 4 | (5) model 5 | (6) model 6 |
|---|---|---|---|---|---|---|
| TR | 1.45155*** (3.05) | | | | | 0.33689 (1.01) |
| CPI | | 4.71027*** (8.62) | | | | 2.75596*** (5.75) |
| BP | | | 1.33767*** (5.72) | | | 0.31341 (1.44) |
| CR | | | | 1.42104*** (12.18) | | 1.07993*** (8.35) |
| TP | | | | | 0.00165* (1.83) | −0.00162** (−2.41) |
| Fiscal expenditure/GDP | | | | | | 0.04868 (0.97) |
| _cons | 5.67517*** (6.13) | −0.02979 (−0.03) | 1.15551 (0.90) | 2.97918*** (4.78) | 7.12566*** (9.27) | −3.57859* (−1.72) |
| N | 118 | 118 | 118 | 118 | 118 | 118 |
| r2 | 0.074 | 0.391 | 0.220 | 0.561 | 0.028 | 0.693 |
| r2_a | 0.07 | 0.39 | 0.21 | 0.56 | 0.02 | 0.68 |

* $p<0.1$, ** $p<0.05$, *** $p<0.01$.

Notably, CPI has a significant positive impact on maximum outcome payments at the 1% level. This supports acceptance of Hypothesis 2 (H2), and proves that CPI has a positive correlation with maximum outcome payment. In this paper, CPI serves as the proxy variable for inflation. Therefore, the empirical finding also indicates that the degree of inflation affects SIB maximum outcome payments. Inflation can cause the cost of capital to rise because bond issuers need to pay higher interest rates to attract investors [37]. This could make SIBs more expensive to issue, affecting their interest rates. In addition, inflation can cause the currency to depreciate, thereby reducing the real purchasing power of future principal and interest payments on bonds. Investors may demand higher interest rates to compensate for this loss, affecting the pricing of SIBs.

## Analysis of bond characteristic factors

Among bond characteristic factors, bond period is not a statistically significant factor. This result refutes Hypothesis 3 (H3) and shows that there is no significant correlation between bond period and maximum outcome payments. Unlike traditional bonds, which are dominated by "finance-first" investors emphasizing liquidity premiums and term risk compensation, the SIB investor structure centers on philanthropic capital and social-mission-driven institutions, which exhibit significant heterogeneity in risk preferences. On one hand, philanthropic capital mitigates market capital's risk exposure by prioritizing initial loss absorption (e.g., a significant portion of investments in the New York SIB case were guaranteed by foundations), thereby diminishing the binding force of term duration on risk pricing [50]. On the other hand, investors may accept below-market returns in exchange for reputational premiums, reducing sensitivity to long-term yield fluctuations [51].

The second characteristic factor, capital raised, has a coefficient of 1.16899 and is significant at the 1% level, which justifies acceptance of Hypothesis 4 (H4). This indicates that the greater the capital raised, the higher the maximum outcome payment of the SIB. Outcome-based SIBs are different from ordinary bonds in that most SIBs have no margin and the public sector transfers financial risk to the investors [52]. If the target results are not achieved, investors face higher risk of losing their investment [53]. SIBs therefore need to formulate the maximum outcome payments for different investors according to the amount of their investment.

For the third characteristic factor, target population, the obtained coefficient is −0.00093 and the statistical result is significant at the 10% level, which supports acceptance of Hypothesis 5 (H5). The coefficient sign indicates a negative correlation of target population with the maximum outcome payment. The reason for this is that SIB payments are related to the amount of the target population that meets the targeted outcome instead of the total number participating in the project. Thus, this result reflects the fact that SIBs pay on outcomes instead of inputs or outputs.

## Discussion

The existing method of formulating SIB maximum outcome payments often focus on the government's goal of reducing fiscal expenditure, while insufficiently considering the interests and risks borne by investors. Formulating maximum outcome payments through an improper method could cause a financing bottleneck and hinder the sustainable advancement of projects. To ensure a reasonable balance between government cost control and investor returns, this study systematically examines the key factors influencing the maximum outcome payments from the perspectives of macroeconomic factors and micro-level bond characteristics. This study selects SIB issued between 2010 and 2024 as the sample and conducts a multiple regression analysis to explore the correlations between these variables and the maximum outcome payments. The results show that inflation, capital raised, and size of target population all exert significant influence on maximum outcome payments. However, the influence of bond period and the rates of treasury bonds was not significant. Given these findings, the study highlights the shortcomings of the current SIB maximum outcome payments pricing methods and provides empirical evidence for improving the current method.

With regard to macro factors, scholars and practitioners have used the risk-free treasury rate as the benchmark interest rate for a long time [23,24]. This paper treated the rates of treasury bonds having the same issuing date and period as the benchmark interest rate of SIBs. However, this work found that the treasury rate does not affect the maximum outcome payments of SIBs. This finding is inconsistent with the current researches that used the risk-free treasury rate as the benchmark interest rate of bonds. Why traditional benchmarks are less relevant is clarified by Theory of Change, namely that SIBs returns are outcome-contingent, not market-driven. Edmiston & Nicholls (2018) argued that the returns of SIBs are entirely contingent on predefined social outcomes (e.g., reducing youth recidivism rates, improving housing stability for the homeless), rather than traditional bond market interest rates or fixed-income mechanisms. This "payment-by-results" framework decouples investor returns from market-driven benchmarks (such as risk-free treasury rate) and directly ties financial rewards to the measurable effectiveness of social service interventions. Besides that, this work identified a positive correlation of inflation with the maximum outcome payment. Inflation can cause the currency to depreciate, thereby reducing the real purchasing power of future principal and interest payments on bonds. Investors may demand higher interest rates to compensate for this loss. This finding is inconsistent with that of Perovic [54], who noted that inflation has a negative impact on government bond yields in Central and Eastern European countries. However, Tekula & Andersen [27] also determined that inflation has a positive effect on government bond yields. Inflation can cause the cost of capital to rise because bond issuers need to pay higher interest rates to attract investors.

In terms of the characteristics of bonds, duration does not affect the maximum outcome payments of SIBs. According to the term structure of interest rates and liquidity preference theory, the longer the maturity of a bond, the higher the interest rate risk and reinvestment risk faced by investors. Bonds with longer duration should be given higher coupon rates; however, the investor composition of SIBs exhibits fundamental heterogeneity compared to conventional bonds. Unlike institutional investors in traditional debt markets, who emphasize liquidity premiums and term risk compensation, SIBs attract philanthropic capital and mission-driven organizations that prioritize social impact over financial returns [50]. In addition, SIBs shorten the actual risk cycle through flexible mechanisms such as phased payments and conditional trigger clauses (e.g., Denmark's GSB project linking only 10% of payments to long-term outcomes), thereby diluting the impact of duration on pricing [55]. The study identified a positive correlation of capital raised with the maximum outcome payment. Relevant studies emphasize that when the issuance scale is large, risk of debt repayment and outcome achievement will increase,

eventually causing bond interest rates to rise [38,42,56]. Accordingly, the investors with higher investments should be given correspondingly higher payments.

The results of this work show that the size of the SIB target population has a negative impact on its maximum outcome payment. According to Theory of Change, SIBs are not paid based on input or output, but rather on outcome [31,33]. The payer thus pays the investors only if the target population achieves the target outcome. It is therefore necessary to properly control target population size and improve the quality of services. Some practitioners and scholars believe that the beneficiaries of SIBs are relatively limited at present [57,58]. However, expanding the scale of SIBs, especially with regard to the number of beneficiaries, does not align with their ultimate goal in principle. That is, we should not measure the influence of SIBs solely based on the number of beneficiaries, but rather by the quality of the services.

Based on the empirical analysis results and the situation discussed above, the following policy implications are put forward:

First, issuers of SIBs should adjust SIB maximum outcome payments according to macroeconomic factors (e.g., inflation) and bond-specific characteristics (e.g., capital raised, target population size). For instance, incorporating inflation-indexed adjustments into outcome payments can protect investor returns against currency depreciation risks. In addition, when selecting the target population, issuers should prioritize quality over quantity. Smaller, well-defined cohorts with high intervention efficacy may yield better outcomes and justify higher payments, which aligns with the negative correlation between target population size and maximum payments observed in this study. Risk-sharing mechanisms should also be clearly defined in SIBs contracts, particularly for large capital raises. Investors require higher payments for larger investments due to heightened credit and repayment risks, necessitating tiered payment structures tied to capital commitments.

Second, governments should require inflation-linked payment adjustment mechanisms to be included in SIB contracts, especially in countries with high economic volatility. Dynamically binding payment terms to inflation indices can effectively hedge against currency depreciation risks and ensure investor confidence during periods of macroeconomic turmoil. However, policy effectiveness depends on a transparent implementation environment. To this end, governments need to simultaneously establish mandatory disclosure standards for SIB core indicators (such as results verification processes and payment trigger conditions) to solve the current information asymmetry dilemma in the market. While consolidating the institutional foundation, governments can further activate market participation through fiscal tools. For example, income tax exemptions can be given to SIB projects that focus on key social issues (such as homelessness and recidivism correction). Tax incentives can offset the additional social risks borne by investors.

Third, investors should prioritize SIBs that explicitly integrate inflation-indexed adjustments into outcome payment structures. Given the significant positive correlation between inflation (proxied by CPI) and maximum outcome payments, contractual clauses linking payments to inflation indices are essential to hedge against currency depreciation risks, especially in volatile economies. The amount of capital raised exhibits a strong positive relationship with maximum payments, reflecting heightened credit and repayment risks for larger issuances. Investors should advocate for tiered payment structures tied to verified milestones, ensuring risk-adjusted returns align with capital exposure. Additionally, the negative correlation between target population size and maximum payments underscores the importance of prioritizing quality over quantity. Smaller, well-defined cohorts with rigorous outcome metrics (e.g., recidivism reduction among high-risk youth) are likely to yield higher risk-adjusted returns compared to diffuse programs targeting large populations.

## Conclusion

The maximum outcome payment is calculated as the sum of service costs and fiscal savings, but often prioritize reducing government costs over investor interests and risks. To find a balance between government cost control and investor returns, this study comprehensively considers the main factors influencing the maximum outcome payments of SIBs from the two aspects of macroeconomic factors and bond characteristics. In addition, current studies on the maximum outcome payments of SIBs have mainly focused on qualitative research, with few relevant quantitative studies. Most are case

studies and are limited to SIBs in two or several countries. Therefore, the available literature is limited in terms of samples and research methods. Distinct from existing qualitative studies, this paper selects all SIBs issued globally as the sample and conducts an empirical study on factors influencing their maximum outcome payments. This analysis revealed that the maximum outcome payments of SIBs are correlated with inflation, capital raised, and size of target population. Consequently, this paper demonstrates that the existing method of pricing SIB maximum outcome payments is unreasonable. To ensure a reasonable balance between government cost control and investor returns, issuers should consider not only fiscal savings but also bond characteristics and macroeconomic factors.

The macro factor found to influence the maximum outcome payments is the inflation. Specifically, inflation has a significant positive correlation with the maximum outcome payment, which validates that inflationary pressures necessitate yield compensation. Inflation can cause the currency to depreciate, thereby reducing the real purchasing power of future principal and interest payments on bonds [37]. Investors may demand higher interest rates to compensate for this loss, affecting the pricing of SIBs. Meanwhile, the bond characteristics found to affect maximum outcome payments are capital raised, and size of target population. The amount of capital raised exhibit a positive correlation with the maximum outcome payment. To compensate for debt repayment risk and outcome achievement risk, the investors with higher investments should be given correspondingly higher payments [42]. Unlike traditional bonds, SIBs transfer financial risk to investors, requiring outcome-based payments scaled to investment amounts to mitigate potential losses [43, 44]. Distinct from capital raised, the size of the target population has a negative correlation with maximum outcome payment. This supports that SIBs pay by outcomes instead of inputs or outputs [9,33]. Service providers should provide high-quality services according to the demands of the target population so that they can achieve the target results and then improve the payments.

Although this study comprehensively considers the main factors influencing the maximum outcome payments of SIBs from the two aspects of macro factors and bond characteristics, there are some limitations as follows. In terms of sample data, the number of SIBs currently issued is small, due to the short development time of SIBs. The limitation provides opportunities for future research and further exploration of the subject. As the SIBs market becomes more mature and improves, the data will continue to be enriched. In the future, relevant research on the factors affecting the maximum outcome payments of SIBs can be further developed to better provide a decision-making reference for issuers, investors, and policymakers.

## Supporting information

**S1 Fig. Research hypothesis.**
(DOC)

**S2 Fig. Regression coefficients plot.**
(DOC)

## Author contributions

**Conceptualization:** Huan Wang, Mengqi Chen, Tingting Fu.

**Data curation:** Huan Wang.

**Methodology:** Huan Wang.

**Project administration:** Huan Wang.

**Supervision:** Naipeng Chao, Jiaxi Chen.

**Validation:** Naipeng Chao, Jiaxi Chen.

**Writing – original draft:** Huan Wang.

**Writing – review & editing:** Huan Wang, Naipeng Chao, Jiaxi Chen.

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
