## [Decision Letter · Decision Letter 0]

Dear Dr. Wang,

Thank you for submitting your manuscript to PLOS ONE. After careful consideration, we feel that it has merit but does not fully meet PLOS ONE’s publication criteria as it currently stands. Therefore, we invite you to submit a revised version of the manuscript that addresses the points raised during the review process.

We look forward to receiving your revised manuscript.

Kind regards,

Vasileios Kallinterakis

Academic Editor

PLOS ONE

 [This work was supported by China Postdoctoral Science Foundation (Certificate Number: 2024M752096).]. 

Additional Editor Comments:

The reviewers have come back with a series of comments, emphasizing room for improvement in the motivation of the paper and its discussion of results. I strongly encourage you to work on addressing their comments.

Reviewers' comments:

Reviewer's Responses to Questions

**Comments to the Author**

1. Is the manuscript technically sound, and do the data support the conclusions?

Reviewer #1: Yes

Reviewer #2: Yes

2. Has the statistical analysis been performed appropriately and rigorously?

Reviewer #1: Yes

Reviewer #2: Yes

3. Have the authors made all data underlying the findings in their manuscript fully available?

Reviewer #1: No

Reviewer #2: Yes

4. Is the manuscript presented in an intelligible fashion and written in standard English?

Reviewer #1: No

Reviewer #2: Yes

Reviewer #1: 1. In this paper, the originality of both new and important information is addressed. However, in the abstract you need to explain why this topic is important by exploiting regulatory, reform and policy issues and developments within the research setting.

2. In the first paragraph of the introduction, the authors must add a reference to the cited figures.

3. The authors must note both the seminal and latest papers in this area.

4. The authors must demonstrate the figures of the descriptive statistics in a paragraph style.

5. Closely link up and cite the papers that you have noted in the literature review for the findings you are presenting in the conclusion section.

6. The authors must revise the citation within the text.

7. Seeking a professional proofreader’s help will be beneficial.

Reviewer #2: This manuscript presents a well-structured empirical study on the macroeconomic and bond-specific factors influencing maximum outcome payments (MOPs) in Social Impact Bonds (SIBs). The authors use a global dataset and conduct a robust statistical analysis—including heteroscedasticity correction, endogeneity control (Heckman model), and robustness tests—which supports the validity of their findings.

The conclusions are well-supported by the data, and the paper contributes meaningfully to both academic and policy discussions in the field of impact finance.

✅ Strengths:

Clear research question and well-motivated hypotheses

Thoughtful use of quantitative methods

Strong relevance to SIB practice and pricing

Addresses a gap in the current literature

Suggestions for Improvement:

Theoretical Integration: The paper uses multiple frameworks (APT, Theory of Change, Public Management), but they are currently treated in parallel. Integrating them into a cohesive conceptual model would improve clarity and coherence.

Variable Justification: The non-significance of Treasury Rate and Bond Period should be discussed more thoroughly in conceptual terms, not only statistically. Consider explaining why these may be less relevant to SIBs than to traditional financial instruments.

Data Transparency: While data availability is sufficient, the reliance on gray literature should be acknowledged with a brief note on quality control and triangulation.

Investor Typology: A brief mention of how findings might differ for “impact-first” vs. “finance-first” investors would add nuance to the results.

Policy Implications: Consider adding a brief subsection outlining practical implications for SIB issuers, governments, and investors.

Figures: A marginal effects plot or regression summary graphic would improve accessibility and highlight key takeaways visually.

Language: The manuscript is largely clear and well-written. Some minor grammatical improvements and clarification of definitions (e.g., of MOP across different country contexts) would improve precision.

Overall, this is a strong and timely manuscript with only moderate revisions necessary to enhance clarity and impact. I support its publication after these refinements.

**Do you want your identity to be public for this peer review?** For information about this choice, including consent withdrawal, please see our Privacy Policy

Reviewer #1: **Yes: ** Dr. Majd Munir Iskandrani

Reviewer #2: **Yes: ** Dr. Lampros Lamprinidis

---

## [Author Response · Author response to Decision Letter 1]

22 May 2025

Dear reviewers,

Thank you very much for your valuable comments on our manuscript titled "Factors Affecting the Maximum Outcome Payments of Social Impact Bonds" (Manuscript ID: PONE-D-25-04221). We greatly appreciate your thoughtful feedback. We have carefully considered all the suggestions and have revised the manuscript accordingly.

Reviewer #1:Dr. Majd Munir Iskandrani

1. In this paper, the originality of both new and important information is addressed. However, in the abstract you need to explain why this topic is important by exploiting regulatory, reform and policy issues and developments within the research setting.

Thanks for the comment. We have revised the abstract as requested to explain the significance of the research topic.

2. In the first paragraph of the introduction, the authors must add a reference to the cited figures.

Thanks for the comment. We have added references to the cited figures in the first paragraph of the introduction. Between the issuance of the first SIB in the UK, in 2010, and June 2024, 295 SIBs had been issued with a total size of about 764m USD according to the Government Outcomes Lab Impact Bond Dataset [2].

3. The authors must note both the seminal and latest papers in this area.

Thanks for the comment. We have noted both the seminal and latest papers in this area. This rapid adoption has sparked intense academic and policy debates, crystallizing around three dominant paradigms: reformist perspectives emphasizing public sector innovation [3, 4], financial models exploring sustainable investment frameworks [5-7], and critical examinations of marketization effects [8-10]. These competing narratives reflect both the transformative potential of SIBs and the implementation challenges they face as they transition from experimental pilots to mainstream policy instruments.

4. The authors must demonstrate the figures of the descriptive statistics in a paragraph style.

Thanks for the comment. We have demonstrated the figures of the descriptive statistics in a paragraph style.

5. Closely link up and cite the papers that you have noted in the literature review for the findings you are presenting in the conclusion section.

Thanks for the comment. We have linked up and cite the papers that we have noted in the literature review for the findings we are presenting in the discussion and conclusion section. The macro factor found to influence the maximum outcome payments is the inflation. Specifically, inflation has a significant positive correlation with the maximum outcome payment, which validates that inflationary pressures necessitate yield compensation. Inflation can cause the currency to depreciate, thereby reducing the real purchasing power of future principal and interest payments on bonds [39]. Investors may demand higher interest rates to compensate for this loss, affecting the pricing of SIBs. Meanwhile, the bond characteristics found to affect maximum outcome payments are capital raised, and size of target population. The amount of capital raised exhibit a positive correlation with the maximum outcome payment. To compensate for debt repayment risk and outcome achievement risk, the investors with higher investments should be given correspondingly higher payments [44]. Unlike traditional bonds, SIBs transfer financial risk to investors, requiring outcome-based payments scaled to investment amounts to mitigate potential losses [45, 46]. Distinct from capital raised, the size of the target population has a negative correlation with maximum outcome payment. This supports that SIBs pay by outcomes instead of inputs or outputs [9, 35]. Service providers should provide high-quality services according to the demands of the target population so that they can achieve the target results and then improve the payments.

6. The authors must revise the citation within the text.

Thanks for the comment. We have revised the citation within the text.

7. Seeking a professional proofreader’s help will be beneficial.

Thanks for the comment. We have improved the text with the help of professional proofreader.

Reviewer #2: Dr. Lampros Lamprinidis

1.Theoretical Integration: The paper uses multiple frameworks (APT, Theory of Change, Public Management), but they are currently treated in parallel. Integrating them into a cohesive conceptual model would improve clarity and coherence.

Thanks for the comment. We have integrated multiple frameworks (APT, Theory of Change, Public Management) into a cohesive conceptual model to systematically explain SIBs' pricing mechanisms. The model establishes macroeconomic variables and bond characteristics from APT as the foundational pricing layer, transforms social value through theory of change's "input-output-outcome" chain, and ultimately balances stakeholder interests via public management theory. These three tiers form a progressive relationship of financial pricing → social adjustment → policy regulation, simultaneously explaining empirical findings while informing policy design, revealing SIBs' unique logic as a market-society-government tripartite coupling mechanism.

2.Variable Justification: The non-significance of Treasury Rate and Bond Period should be discussed more thoroughly in conceptual terms, not only statistically. Consider explaining why these may be less relevant to SIBs than to traditional financial instruments.

Thanks for the comment. We have discussed the non-significance of Treasury Rate and Bond Period more thoroughly in conceptual terms. We have explained why these may be less relevant to SIBs than to traditional financial instruments. The multiple regression revealed that the treasury rate, as the benchmark interest rate, is not a statistically significant factor. This result refutes Hypothesis 1 (H1) and shows that there is no significant correlation between the treasury rate and maximum outcome payments. The fundamental distinction between SIBs and traditional bonds lies in their risk pricing logic: Traditional bond returns depend on issuer creditworthiness and market interest rate fluctuations, whereas SIBs tie returns entirely to third-party-verified achievement of social outcomes (e.g., crime rate reduction or employment rate improvement) through "outcome-based payment" contracts, forming a "results-driven" pricing model [50]. The core reason for the insignificance of treasury rates in this framework is that SIBs reconstruct the value assessment system through risk transfer. Their payment models are grounded in the causal effects of social interventions (such as net impacts validated by randomized controlled trials) rather than financial market interest rate parameters [51]. Concurrently, domination of the investor structure by philanthropic capital and the social reputation premium compensation mechanism [52] further diminish the benchmark role of risk-free rates in pricing.

Among bond characteristic factors, bond period is not a statistically significant factor. This result refutes Hypothesis 3 (H3) and shows that there is no significant correlation between bond period and maximum outcome payments. Unlike traditional bonds, which are dominated by “finance-first” investors emphasizing liquidity premiums and term risk compensation, the SIB investor structure centers on philanthropic capital and social-mission-driven institutions, which exhibit significant heterogeneity in risk preferences. On one hand, philanthropic capital mitigates market capital’s risk exposure by prioritizing initial loss absorption (e.g., a significant portion of investments in the New York SIB case were guaranteed by foundations), thereby diminishing the binding force of term duration on risk pricing [52]. On the other hand, investors may accept below-market returns in exchange for reputational premiums, reducing sensitivity to long-term yield fluctuations [53].

3.Data Transparency: While data availability is sufficient, the reliance on gray literature should be acknowledged with a brief note on quality control and triangulation.

Thanks for the comment. We have explained the gray literature on quality control and triangulation. Data on the maximum outcome payment, bond period, capital raised and size of target population were obtained from the Government Outcomes Lab created as a partnership between the University of Oxford and the UK Government. Missing data were supplemented from the gray literature, which includes the website of each SIB and its report. To ensure data quality and comparability, all gray literature sources were subject to a triangulation process involving cross-verification with multiple independent sources, prioritization of government or evaluator-backed documents, and consistency checks.

4.Investor Typology: A brief mention of how findings might differ for “impact-first” vs. “finance-first” investors would add nuance to the results.

Thanks for the comment. We have a brief mention of how findings might differ for “impact-first” vs. “finance-first” investors in the results analysis section. Unlike traditional bonds, which are dominated by “finance-first” investors emphasizing liquidity premiums and term risk compensation, SIBs’ investor structure centers on philanthropic capital and social mission-driven institutions (“impact-first”s investors), exhibiting significant heterogeneity in risk preferences. On one hand, philanthropic capital mitigates market capital’s risk exposure by prioritizing initial loss absorption (e.g., a significant portion of investments in the New York SIB case were guaranteed by foundations), thereby diminishing the binding force of term duration on risk pricing. On the other hand, investors may accept below-market returns in exchange for reputational premiums, reducing sensitivity to long-term yield fluctuations.

5.Policy Implications: Consider adding a brief subsection outlining practical implications for SIB issuers, governments, and investors.

Thanks for the comment. We have adding a brief subsection in discussion section outlining practical implications for SIB issuers, governments, and investors.Based on the empirical analysis results and the situation discussed above, the following policy implications are put forward:

First, issuers of SIBs should adjust SIB maximum outcome payments according to macroeconomic factors (e.g., inflation) and bond-specific characteristics (e.g., capital raised, target population size). For instance, incorporating inflation-indexed adjustments into outcome payments can protect investor returns against currency depreciation risks. In addition, when selecting the target population, issuers should prioritize quality over quantity. Smaller, well-defined cohorts with high intervention efficacy may yield better outcomes and justify higher payments, which aligns with the negative correlation between target population size and maximum payments observed in this study. Risk-sharing mechanisms should also be clearly defined in SIBs contracts, particularly for large capital raises. Investors require higher payments for larger investments due to heightened credit and repayment risks, necessitating tiered payment structures tied to capital commitments.

Second, governments should require inflation-linked payment adjustment mechanisms to be included in SIB contracts, especially in countries with high economic volatility. Dynamically binding payment terms to inflation indices can effectively hedge against currency depreciation risks and ensure investor confidence during periods of macroeconomic turmoil. However, policy effectiveness depends on a transparent implementation environment. To this end, governments need to simultaneously establish mandatory disclosure standards for SIB core indicators (such as results verification processes and payment trigger conditions) to solve the current information asymmetry dilemma in the market. While consolidating the institutional foundation, governments can further activate market participation through fiscal tools. For example, income tax exemptions can be given to SIB projects that focus on key social issues (such as homelessness and recidivism correction). Tax incentives can offset the additional social risks borne by investors.

Third, investors should prioritize SIBs that explicitly integrate inflation-indexed adjustments into outcome payment structures. Given the significant positive correlation between inflation (proxied by CPI) and maximum outcome payments, contractual clauses linking payments to inflation indices are essential to hedge against currency depreciation risks, especially in volatile economies. The amount of capital raised exhibits a strong positive relationship with maximum payments, reflecting heightened credit and repayment risks for larger issuances. Investors should advocate for tiered payment structures tied to verified milestones, ensuring risk-adjusted returns align with capital exposure. Additionally, the negative correlation between target population size and maximum payments underscores the importance of prioritizing quality over quantity. Smaller, well-defined cohorts with rigorous outcome metrics (e.g., recidivism reduction among high-risk youth) are likely to yield higher risk-adjusted returns compared to diffuse programs targeting large populations.

6.Figures: A marginal effects plot or regression summary graphic would improve accessibility and highlight key takeaways visually.

Thanks for the comment. We have added a regression coefficients plot as supporting information. The regression coefficients plot for determinants of maximum outcome payments is shown in S2 Fig. The results are consistent with the regression findings, indicating that CPI, CR, and TP have a significant effect on maximum outcome payments.

7.Language: The manuscript is largely clear and well-written. Some minor grammatical improvements and clarification of definitions (e.g., of MOP across different country contexts) would improve precision.

Thanks for the comment. We have improved grammar with the help of professional proofreader and clarified the definitions of maximum outcome payments in the introduction section. The maximum outcome payment refers to the highest possible amount that the outcome payer commits to pay to investors if all agreed-upon outcomes are fully achieved [5]. This cap serves both as a risk containment measure for the payer and an outcome-based incentive for investors. However, the definition and application of maximum outcome payment may vary across projects and policy objectives. For example, the maximum outcome payment in the UK's HMP Peterborough SIB was explicitly tied to reductions in recidivism among short-sentenced prisoners, with predefined thresholds and payment rates [12]. Conversely, in the Utah High Quality Preschool Program SIB in the United States, the payment was based on the number of children avoiding special education placement, with a fixed per-child payment that accrued up to a maximum cap [13].

---

## [Decision Letter · Decision Letter 1]

Factors affecting the maximum outcome payments of social impact bonds

PONE-D-25-04221R1

Dear Dr. Wang,

We’re pleased to inform you that your manuscript has been judged scientifically suitable for publication and will be formally accepted for publication once it meets all outstanding technical requirements.

Kind regards,

Vasileios Kallinterakis

Academic Editor

PLOS ONE

Additional Editor Comments (optional):

The reviewer is happy with the way you have addressed their comments and recommends acceptance.

Reviewers' comments:

Reviewer's Responses to Questions

**Comments to the Author**

Reviewer #2: All comments have been addressed

2. Is the manuscript technically sound, and do the data support the conclusions?

Reviewer #2: Yes

3. Has the statistical analysis been performed appropriately and rigorously?

Reviewer #2: Yes

4. Have the authors made all data underlying the findings in their manuscript fully available?

Reviewer #2: Yes

5. Is the manuscript presented in an intelligible fashion and written in standard English?

Reviewer #2: Yes

Reviewer #2: Thank you for your detailed and thoughtful revision. The authors have adequately addressed all my previous comments and significantly improved the manuscript in clarity, coherence, and rigor.

The integration of the three theoretical frameworks—Arbitrage Pricing Theory (APT), Theory of Change, and Public Management—is now conceptually cohesive and well-articulated. The multi-layered structure (financial pricing → social adjustment → policy regulation) meaningfully supports the empirical findings and provides a clearer lens for understanding SIBs’ pricing mechanisms. This theoretical synthesis enhances the paper’s conceptual depth and makes it more informative for both academic and policy audiences.

The lack of statistical significance for Treasury Rate and Bond Period is now properly contextualized. The authors present a compelling explanation rooted in the distinct outcome-contingent logic of SIBs, which diverge from conventional risk-pricing frameworks of traditional bonds. The discussion on investor typologies (impact-first vs finance-first) and their effect on risk preferences further strengthens the argument.

Data sourcing and gray literature triangulation procedures are now transparently presented, which increases the credibility and replicability of the analysis. The inclusion of a regression coefficients plot enhances the accessibility of key results. Language, definitions (e.g., Maximum Outcome Payment), and structure have been professionally improved.

Finally, the added policy implications section is both practical and insightful, offering actionable recommendations for SIB issuers, government stakeholders, and investors. In my view, this revised version is technically sound, methodologically rigorous, well-written, and ready for publication.

**Do you want your identity to be public for this peer review?** For information about this choice, including consent withdrawal, please see our Privacy Policy

Reviewer #2: **Yes: ** Lampros Lamprinidis, PhD

---

## [Editor Report · Acceptance letter]

PONE-D-25-04221R1

PLOS ONE

Dear Dr. Wang,

I'm pleased to inform you that your manuscript has been deemed suitable for publication in PLOS ONE. Congratulations! Your manuscript is now being handed over to our production team.

Kind regards,

on behalf of

Dr. Vasileios Kallinterakis

Academic Editor

PLOS ONE